# Factors associated with implant survival following total hip replacement surgery: A registry study of data from the National Joint Registry of England, Wales, Northern Ireland and the Isle of Man

Jonathan Thomas Evans[1]*, Ashley William Blom[1,2], Andrew John Timperley[3,4], Paul Dieppe[5], Matthew James Wilson[3], Adrian Sayers[1‡], Michael Richard Whitehouse[1,2‡]

1 Musculoskeletal Research Unit, Translational Health Sciences, Bristol Medical School, Bristol, United Kingdom, 2 National Institute for Health Research Bristol Biomedical Research Centre, University Hospitals Bristol NHS Foundation Trust and University of Bristol, United Kingdom, 3 Princess Elizabeth Orthopaedic Centre, Royal Devon & Exeter NHS Foundation Trust, United Kingdom, 4 College of Engineering, Mathematics and Physical Sciences, University of Exeter, Exeter, United Kingdom, 5 The University of Exeter Medical School, University of Exeter, Exeter, United Kingdom

‡ These authors are joint senior authors on this work.
* j.t.evans@bristol.ac.uk

**Data Availability Statement:** Data cannot be shared publicly. Data are available from the NJR

## Abstract

### Background

Nearly 100,000 people underwent total hip replacement (THR) in the United Kingdom in 2018, and most can expect it to last at least 25 years. However, some THRs fail and require revision surgery, which results in worse outcomes for the patient and is costly to the health service. Variation in the survival of THR implants has been observed between units and reducing this unwarranted variation is one focus of the "Getting it Right First Time" (GIRFT) program in the UK. We aimed to investigate whether the statistically improved implant survival of THRs in a high-performing unit is associated with the implants used or other factors at that unit, such as surgical skill.

### Methods and findings

We analyzed a national, mandatory, prospective, cohort study (National Joint Registry for England, Wales, Northern Ireland and the Isle of Man [NJR]) of all THRs performed in England and Wales. We included the 664,761 patients with records in the NJR who have received a stemmed primary THR between 1 April 2003 and 31 December 2017 in one of 461 hospitals, with osteoarthritis as the only indication. The exposure was the unit (hospital) in which the THR was implanted. We compared survival of THRs implanted in the "exemplar" unit with THRs implanted anywhere else in the registry. The outcome was revision surgery of any part of the THR construct for any reason. Net failure was calculated using

research subcommittee researchers who meet the criteria for access to confidential data. Access to the data used in this study can be requested via njrresearch@hqip.org.uk. Full details of how to request NJR data for research can be found at: http://www.njrcentre.org.uk/njrcentre/Research/Research-requests.

**Funding:** This study was supported by the NIHR Biomedical Research Centre at University Hospitals Bristol and Weston NHS Foundation Trust and the University of Bristol. JTE was supported by the joint National Joint Registry of England, Wales, Northern Ireland and the Isle of Man and Royal College of Surgeons of England Fellowship. AS was supported by a MRC strategic skills fellowship: MRC Fellowship MR/L01226X/1 The funders had no role in study design, data collection and analysis, decision to publish, or preparation of the manuscript.

**Competing interests:** I have read the journal's policy and the authors of this manuscript have the following competing interests: AB and MRW have received a research grant from Stryker for an unrelated knee replacement trial, MRW's institution has been paid for work he has done developing and delivering educational presentations for DePuy and Heraeus. MJW and AJT receive institutional support from Stryker for research into and royalties relating to the Exeter hip replacement system from Stryker; there are no other relationships or activities that could appear to have influenced the submitted work.

**Abbreviations:** ASA, American Society of Anesthesiology; FPSA, flexible parametric survival analysis; GIRFT, Getting it Right First Time; HQIP, Healthcare Quality Improvement Partnership; NHS, National Health Service; NJR, National Joint Registry for England, Wales, Northern Ireland and the Isle of Man; RD&E, Royal Devon & Exeter NHS Foundation Trust; RMST, restricted mean survival time; THR, total hip replacement.

Kaplan–Meier estimates, and adjusted analyses employed flexible parametric survival analysis.

The mean age of patients contributing to our analyses was 69.9 years (SD 10.1), and 61.1% were female. Crude analyses including all THRs demonstrated better implant survival at the exemplar unit with an all-cause construct failure of 1.7% (95% CI 1.3–2.3) compared with 2.9% (95% CI 2.8–3.0) in the rest of the country after 13.9 years (log-rank test $P < 0.001$). The same was seen in analyses adjusted for age, sex, and American Society of Anesthesiology (ASA) score (difference in restricted mean survival time 0.12 years [95% CI 0.07–0.16; $P < 0.001$]). Adjusted analyses restricted to the same implants as the exemplar unit show no demonstrable difference in restricted mean survival time between groups after 13.9 years ($P = 0.34$).

A limitation is that this study is observational and conclusions regarding causality cannot be inferred. Our outcome is revision surgery, and although important, we recognize it is not the only marker of success of a THR.

## Conclusions

Our results suggest that the "better than expected" implant survival results of this exemplar center are associated with implant choice. The survival results may be replicated by adopting key treatment decisions, such as implant selection. These decisions are easier to replicate than technical skills or system factors.

## Author summary

### Why was this study done?

- In general, total hip replacement (THR) is safe and effective at reducing pain and restoring mobility to people with end-stage arthritis of the hip.

- In England and Wales, in 2017, over 822 different types of hip replacement were used, and different brands of hip replacement have been shown to have varying survival rates at different follow-up timepoints. Reducing variation in outcomes following surgery is an important aim of the National Health Service (NHS) in England and Wales.

- A national database of all hip replacements in England and Wales (the National Joint Registry) has shown variation in survival rates between different hospitals, and a few hospitals are highlighted by the database for having better survival rates than the others.

### What did the researchers do and find?

- One of the hospitals with better survival rates for hip replacements than the others uses only one type of hip replacement for all patients.

- We compared the survival of THRs implanted in this one hospital to THRs implanted anywhere else in the country to look for factors that are associated with improved survival.

- When this hospital was compared with everyone else using the same hip replacement, after taking the patients' age, sex, and general health into account, they no longer had better results than anyone else.

### What do these findings mean?

- These findings suggest that the better results seen in this one hospital are not associated with the skill of the surgeon or the setup of the hospital but are associated with the choice of hip replacement.

- Future studies are needed to determine whether this is also the case across other brands of hip replacement and to determine whether the choice of implant is similarly associated with implant survival across other specialties.

## Introduction

Total hip replacement (THR) is one of the most successful operations of our time with nearly 100,000 performed in England, Wales, Northern Ireland, and the Isle of Man in 2017 [1, 2]. They have been shown in general to last about 25 years, but despite this, there is still variation in the survival of implants across the UK [2, 3]. THR components may require changing (via revision surgery) for one of several reasons including infection, wear, loosening, fracture, or instability [2]. The need for future revision surgery can be influenced by preoperative patient factors, implant factors, and surgical factors [4, 5]. It has previously been demonstrated that THR revisions are not as effective in improving pain and function as the primary operation, have a high chance of further revision, and are costly to the health service, as well as resulting in exposure of patients to the additional pain and inconvenience of another operation [2, 6, 7]. Although implant survival is not the only marker of success [8], the cumulative probability of revision of THRs is a readily available outcome measure because of the National Joint Registry for England, Wales, Northern Ireland and the Isle of Man (NJR) a mandatory, national database [9].

The NJR has collected data since 2003 and at the time of writing contains in excess of 1 million records of primary THRs. In 2017, the NJR identified at least 415 different units (hospitals) performing THRs using at least 822 different combinations of femoral stem and acetabular socket [10, 11] and is thought to capture over 95% of primary hip and knee operations and 90% of revisions [2]. Every year, the NJR annual report lists units in which either a higher or lower than expected rate of revision has been observed over the preceding years. The units with a "better than expected" (above the 99.7% confidence limit) revision estimate may offer an opportunity to learn from good practice and potentially reduce variation between units.

The importance of reducing unwarranted variation across the whole National Health Service (NHS) has been highlighted by the recent work of the "Getting it Right First Time" (GIRFT) program [12]. This review has previously highlighted variation in adult elective orthopedic services and makes clear the requirement to learn from good performance [13].

Investigating what may lead one unit to demonstrate better results than others is challenging because of the differing patient populations as well as issues with potential selection bias. Patients may receive different types of implants based on factors such as age and sex, and as a

result, outcomes are difficult to interpret. One unit in the NJR, the Royal Devon & Exeter NHS Foundation Trust (RD&E), has been repeatedly identified as having "better than expected" survival outcomes and is widely known for using only one femoral stem (the Exeter V40 femoral stem) in all routine primary THRs regardless of patient factors, thus removing or reducing selection bias [2]. This offers an opportunity to investigate whether "better than expected" survival results observed within this unit were due to a unit effect or because of the implants used.

## Methods

We aimed to compare the cumulative revision estimates between the RD&E and the rest of the country to investigate whether "better than expected" outcomes were due to the implants used or because of other unit factors.

The NJR is a mandatory national audit of joint replacement activity. After gaining written consent, operations are reported to the NJR by the healthcare provider at the time of surgery. The dataset consisted of 981,269 linked primary THRs performed in England and Wales between 1 April 2003 and 31 December 2017 with consent for data linkage. Data were censored either by death or administratively on 31 December 2017. After exclusion of THRs with incomplete or inconsistent data or using metal-on-metal bearings, we were left with 664,761 primary THRs, in which osteoarthritis was the only indication for THR. Reasons for exclusion at each stage are shown in Fig 1.

### Statistical analysis

Statistical analysis was performed with Stata 15 (Stata Statistical Software: Release 15. College Station, TX: StataCorp LLC, https://www.stata.com/). The exposure of interest was the unit in which the THR was performed, and the 2 groups were THRs performed at the RD&E and THRs performed in any other unit. The choice of the RD&E as the exposure group (rather than any of the other units with "better than expected" survival results) is due to a lack of

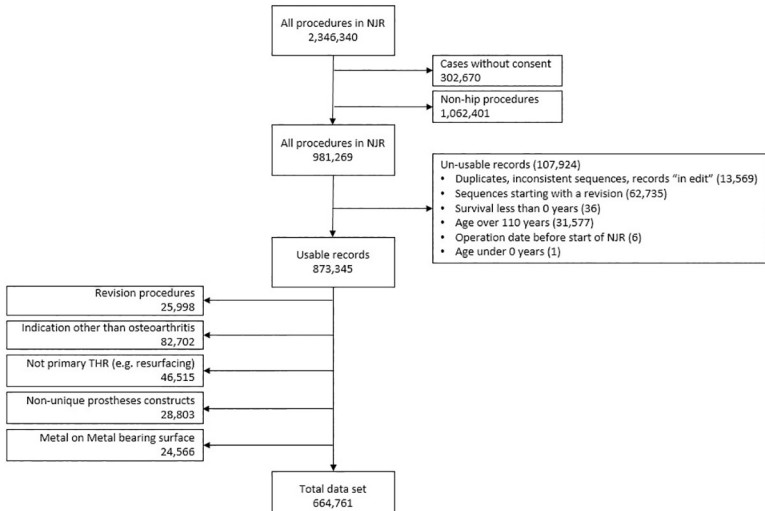

**Fig 1. Reasons for exclusion from analyses.** A sequence is the order of operations recorded in the NJR for any patient. All complete records will start with a primary operation. If a sequence starts with a revision, the primary was performed before the NJR, outside the geographical coverage of the NJR, or data were not submitted to the NJR. THR, total hip replacement; NJR, National Joint Registry for England, Wales, Northern Ireland and the Isle of Man.

selection bias, in that every patient in this unit receives the same femoral stem, regardless of age, sex, or indication. This lack of selection bias is unique to this unit.

The outcome of interest was revision of any part of the THR for any reason. All-cause revision was defined using the NJR definition as the addition, removal or modification of any part of the construct [2]. The study population was all THRs implanted in the NJR; subgroup analysis was performed for THRs using any type of cemented stem (hybrid or all-cemented constructs) as well as THRs using the Exeter V40 femoral stem (hybrid or all-cemented constructs), the stem used by the RD&E.

Unadjusted survival estimates were calculated using the Kaplan–Meier (KM) method for all included THRs, stratified by the exposure of interest [14]. Flexible parametric survival analysis (FPSA), as described by Royston and Parmar, was used to look for time varying effects in the 2 exposure groups by plotting time-dependent against proportional hazards models [15]. FPSA models were then used to compare THRs performed at the RD&E to those performed in any other unit, having adjusted for age, sex, and American Society of Anesthesiology (ASA) score at time of surgery and allowing for time varying effects. FPSA models were assessed visually for goodness of fit against KM curves, in THRs using the Exeter V40 femoral stem. FPSA modeling offers an advantage over more traditional semiparametric techniques, such as Cox regression, because in addition to allowing effects to vary with time via cubic splines, they allow us to estimate a baseline hazard function. This baseline hazard allows the estimation of absolute effects, such as survival, for both groups given certain values of covariates, rather than simply an estimate of the relative effect between the 2 groups (hazard ratio) as is given in Cox regression. Graphs comparing revision estimates between the 2 exposure groups were fitted to models for a 68-year-old female patient, to reflect the median age and most common sex receiving primary THRs in the NJR. Restricted mean survival times were calculated using the standardized survival package "stpm2_standsurv" [16].

Data were censored either by death, or administratively on 31 December 2017. THRs with incomplete or inconsistent data or using metal-on-metal bearings (previously shown to demonstrate poorer survival [17]) were excluded, and cases were included only where osteoarthritis was the sole indication for THR. Reasons for exclusion at each stage are shown in Fig 1.

## Sensitivity analyses

Other potential confounders were considered with a priori knowledge and focusing on variables that were determined before the choice of implant was made, rather than those potentially related to implant choice and thus potentially mediators (e.g., surgical approach and anesthetic). Socioeconomic status (SES) and body mass index (BMI) may also be important potential confounders. Socioeconomic status was assessed using deciles of the Index of Multiple Deprivation (IMD) organized by Lower Layer Super Output Areas (LSOAs), and BMI was treated as a categorical variable using World Health Organization categories (<18.5, 18.5–24.9, 25–29.9, 30–34.9, 35–39.9, and >40 kg/m$^2$). S1 Table shows the distribution of BMI across the strata for the overall cohort as well as the 2 exposure groups.

Cumulative revision estimates were explored restricting analyses to only THRs using the same implant combinations as the exemplar center. Construct survival of THRs using the 5 most implanted cemented stems were explored to determine whether the similar results could be achieved with other commonly used implants within the same type of construct fixation.

**Missing data.** Cases missing data on potential confounders (age, sex, ASA score, BMI, or socioeconomic status) were retained in analyses that were not using that specific model as a covariate. A table detailing the distribution of missing data on these covariates can be seen in S2 Table. Data regarding SES were only available for patients operated in England, and BMI

was missing in 30.1% of cases. Complete case analysis models including these variables were completed as sensitivity analyses as multiple imputation of these data may introduce bias if they are not missing truly at random [18].

### Patient and public involvement

Patients were involved in the design of this study through the Patient Experience in Research (PEPR) group at the Musculoskeletal Research Unit, University of Bristol [19], and in the NJR Research Sub-committee who provided authority for this study. The same groups will be involved in the dissemination of results. The choice of outcome of interest (all-cause revision rather than revision for specific indications) was guided by the PEPR group.

### Planning of analyses

The analysis plan was made prior to the start of all analyses and agreed on among co-authors. No data-driven changes to the analysis plan were made.

This study is reported as per the Strengthening the Reporting of Observational Studies in Epidemiology (STROBE) guideline (S1 STROBE Checklist). Approval for this study was granted by the NJR Research Sub-committee (reference RSC2017/15). Written consent was granted by patients for inclusion of their data and its use in research within the NJR.

## Results

After exclusions, we were left with 664,761 primary THRs for analysis. The maximum follow-up in the exemplar center group was 13.9 years and was 14.2 years in all other units. The demographics and distribution of the THRs in each group can be seen in Table 1. Of the 6,230 cases performed at the RD&E, there were 83 different recorded "lead" surgeons who performed a range from one THR to 992 THRs included in the study dataset. A total of 68.1% of

**Table 1. Demographics and distribution of included total hip replacements.**

| | | All total hip replacements | | | | Constructs using a cemented stem | | | | Constructs using the same femoral stem as the exemplar center | | | |
|---|---|---|---|---|---|---|---|---|---|---|---|---|---|
| | | Operated on at the exemplar center | | Not operated on at the exemplar center | | Operated on at the exemplar center | | Not operated on at the exemplar center | | Operated on at the exemplar center | | Not operated on at the exemplar center | |
| Total, n | | 6,230 | | 658,531 | | 6,228 | | 379,691 | | 6,227 | | 228,814 | |
| Female, n (%) | | 3,621 | (58.1) | 402,406 | (61.1) | 3,619 | (58.1) | 245,891 | (64.8) | 3,619 | (58.1) | 146,219 | (63.9) |
| Mean age, SD | | 70.2 | (10.6) | 69.9 | (10.1) | 70.2 | (10.6) | 72.5 | (9.2) | 70.2 | (10.6) | 72.1 | (9.3) |
| Mean body mass index, kg/m$^2$ (SD) | | 28.6 | (5.2) | 28.7 | (5.2) | 28.6 | (5.2) | 28.4 | (5.1) | 28.6 | (5.2) | 28.5 | (5.1) |
| Posterior approach, n (%) | | 5,553 | (89.1) | 377,802 | (57.4) | 5,552 | (89.1) | 208,652 | (55.0) | 5,551 | (89.1) | 136,090 | (59.5) |
| American Society of Anesthesiologists score, n (%) | I | 963 | (15.5) | 98,212 | (14.9) | 963 | (15.5) | 48,189 | (12.7) | 963 | (15.5) | 29,987 | (13.1) |
| | II | 4,499 | (72.2) | 462,006 | (70.2) | 4,497 | (72.2) | 265,746 | (70.0) | 4,496 | (72.2) | 159,682 | (69.8) |
| | III | 756 | (12.1) | 95,507 | (14.5) | 756 | (12.1) | 63,868 | (16.8) | 756 | (12.1) | 38,050 | (16.6) |
| | IV & V | 12 | (0.2) | 2,806 | (0.4) | 12 | (0.2) | 1,888 | (0.5) | 12 | (0.2) | 1,095 | (0.5) |
| National Health Service funded, n (%) | | 6,123 | (98.3) | 552,907 | (84.0) | 6,121 | (98.3) | 317,950 | (83.7) | 6,120 | (98.3) | 193,566 | (84.6) |
| Consultant as operating surgeon, n (%) | | 3,004 | (48.2) | 546,315 | (83.0) | 3,002 | (48.2) | 302,394 | (79.6) | 3,001 | (48.2) | 184,886 | (80.8) |
| Cemented acetabulum, n (%) | | 4,927 | (79.1) | 263,055 | (39.9) | 4,927 | (79.1) | 244,644 | (64.4) | 4,926 | (79.1) | 147,578 | (64.5) |

Table demonstrating the demographics and distribution of all total hip replacements included in this study broken down by exposure groups and sensitivity analysis subgroups.

THRs were performed by 1 of 6 surgeons. The "lead" surgeon may be a consultant (attending), fellow, or higher specialist trainee (resident) operating under the supervision of a consultant.

## Crude analyses

The crude 10-year cumulative revision estimate of all THRs implanted at the RD&E was 1.7% (95% CI 1.3–2.3). In all other units, the 10-year cumulative revision estimate for all THRs was 2.9% (95% CI 2.8–3.0; log-rank test $P < 0.001$); for just THRs using cemented stems, it was 2.6% (95% CI 2.5–2.7; log-rank test $P = 0.007$), and for just THRs using the Exeter V40 femoral stem, it was 2.3% (95% CI 2.2–2.4) (log-rank test $P = 0.05$). Net revision estimates calculated using 1-Kaplan–Meier curves can be seen in Fig 2; the number of hips at risk at all time points for all analyses can be seen in S3 Table.

## Adjusted analyses

A FPSA model was fitted for all THRs using the Exeter V40 femoral stem and showed excellent "goodness of fit" (S1 Fig). Comparison of time-dependent and proportional hazards models suggested the time-dependent model showed better fit (S2 Fig). After adjustment for age, sex, and ASA and allowing for time varying effects, the relative revision estimates of each subgroup of THRs (THRs using a cemented stem or THRs using the Exeter V40 femoral stem) modeled for a 68-year-old, female patient, are shown in Fig 3A, Fig 3B and Fig 3C.

The femoral stem was paired with 9 different acetabular components in the RD&E, and 99% of these THRs used 1 of only 3 cups. In other units, the Exeter V40 femoral stem was paired with 111 different acetabular components. Fig 3D shows a comparison of the 2 groups if analyses are restricted to stem/cup combinations used at the RD&E. This restricted analysis compares 6,227 performed in the RD&E with 148,295 THRs performed elsewhere. After 13.9 years, there is a discrepancy in restricted mean survival time (RMST) of 0.02 years (95% CI −0.02 to 0.07; $P = 0.33$). A $P$ value of 0.33 suggests there is little or no evidence of any difference in survival of THRs after 13.9 years between those implanted in the RD&E compared with

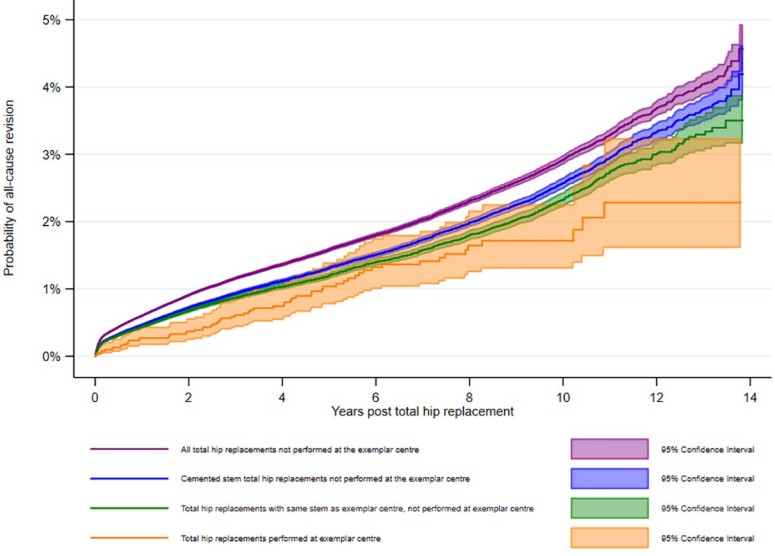

**Fig 2. Unadjusted 1-Kaplan–Meier revision estimates of total hip replacements in each subgroup.** Comparison of the all-cause construct revision estimates of total hip replacements performed in the exemplar center compared with those performed in all other hospitals in the National Joint Registry.

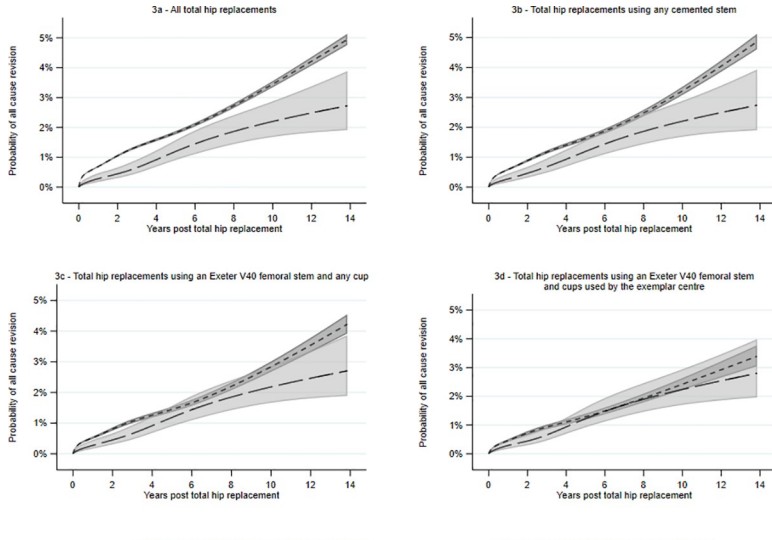

**Fig 3. FPSA adjusted for age, sex, and American Society of Anesthesiology score.** Results presented for a 68-year-old female patient with an American Society of Anesthesiology score of 2. FPSA, flexible parametric survival analysis.

elsewhere when the same implants were used. Fig 4 shows how the difference in RMST between the 2 subgroups changes over time. RMST is reported in years and estimates the difference in life expectancy of the THR between the 2 exposure groups, i.e., the extra time a THR lasts because it was implanted in the RD&E. The difference in RMST changes slightly over time, most notably at 2 time points, 3 years and at 10 years. This may reflect particular modes of failure such as loosening of cups at 10 years, potentially due to cementation technique. It should be noted that for the majority of time reported, the confidence intervals cross the null

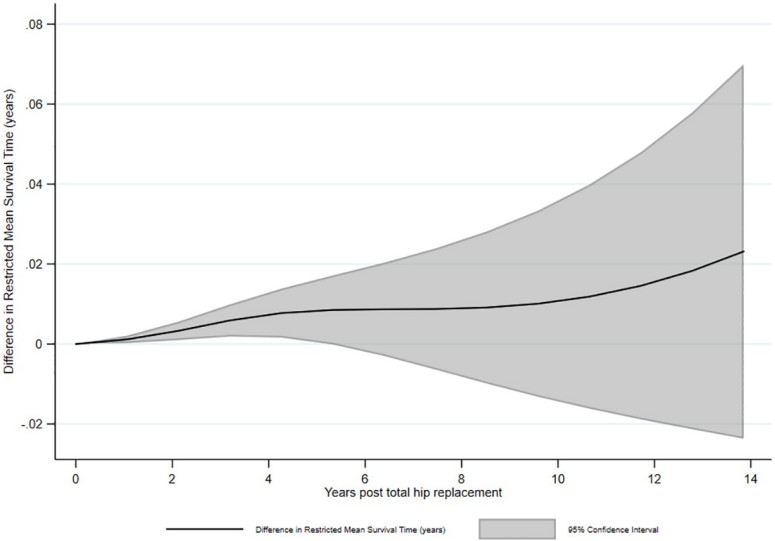

**Fig 4. Difference in RMST between total hip replacements performed at the RD&E and in all other hospitals combined when using the same implants as those used at the RD&E.** A demonstration of how the difference in RMST varies between the 2 exposure groups using the flexible parametric model adjusted for age, sex, and American Society of Anesthesiologists score and allowing for time varying effects. RMST, restricted mean survival time; RD&E, Royal Devon & Exeter.

value of 0. Graphically, it appears that for a short period there may be a transient center effect; however, analysis of the entire period shows no such center effect.

## Sensitivity analyses

A complete case analysis including socioeconomic status in the model excludes 16 cases from the RD&E (0.3%) and 8,569 cases performed elsewhere (5.8%). The results of this model are very similar to that described previously (S3 Fig). Complete case analysis with BMI included in the model, again, shows roughly similar results; however, given the much higher proportion of missing data, the CIs are wider (S4 Fig).

Analysis of the all-cause construct survival of THRs using the 5 most commonly implanted cemented stems across the NJR to date, shows that other stems may achieve comparable performance to the Exeter V40 femoral stem, but this is not true of all stems with the same mode of fixation (S5 Fig).

## Discussion

After 13.9 years, both crude and adjusted cumulative revision estimates showed better implant survival when THRs were performed at the RD&E compared with elsewhere in the country. In analyses adjusted for age, sex, and ASA score, these differences attenuated after restricting to only cemented implants and disappeared when only THRs using the same implant combinations as the RD&E were analyzed. This suggests that implant choice is responsible for the "better than expected" results at the RD&E and not unit (or surgeon) factors.

We are unaware of any studies to date investigating the reasons why 1 unit achieves better THR survival than others. This study suggests that when attempting to improve implant survivorship, units performing THR, particularly those with "lower than expected" implant survival, should focus attention on choice of implant rather than other factors. The use of implants without evidence of good long-term survival should be limited to well-controlled and monitored studies or experiments. Although this study has focused on 1 single femoral stem (the Exeter V40 femoral stem), we believe that the observed high survival would be reproducible with other well performing implants. Previous work by Deere and colleagues has compared the survival of implant combinations after 10 years and provides a reference to demonstrate other implant combinations with low revision rates [5]. The NJR annual report provides a list of units with "better-than-expected" survival results as well as survival estimates for individual stem/cup combinations and can act as a reference document to units wishing to review their implant selection. These findings are of relevance to surgeons, commissioners, and most importantly, patients when deciding whether to, where, and when to have a THR. Patients should be encouraged to ask surgeons about the long-term survival evidence for the implant they plan to use.

The strength of this study stems from the high number of patients included and the use of a linked, national database with high capture of revision procedures. The lack of selection bias in choice of femoral stem at the RD&E (our reference unit) is another strength. The data in this study are, however, observational, and conclusions regarding causality must be interpreted with caution. Our outcome is revision surgery, and although important, it is not the only marker of success of a THR. Patient reported outcomes such as pain and function have not been assessed and patients may have been unsuitable or unwilling to undergo a revision operation and as such a failure may have been misclassified as a success. We made no attempt to restrict by bearing surfaces in this study, which may be a contributing factor in the longevity of a THR; this would, however, have created several subgroups, which we wished to avoid so we could maintain sample size. We would expect the complexity of cases to be generally

representative of the UK population given the NHS referral system and have additionally made attempts to adjust for potential confounders; however, there may still be some residual confounding for variables with incomplete data or not captured by the dataset. There are likely to be sequential hip replacements performed on different sides within the same patient included in this study. We treated each hip as an individual case. There is a risk of failure of one THR leading to subsequent failure of the other side in cases of infection; however, given that this is also the case from other joint replacements (e.g., knee) or from other conditions leading to a higher propensity to infection, we felt this risk was negligible and therefore did not exclude these from analyses. The use of complete case analysis over multiple imputation for handling missing covariates was also a potential weakness and may result in a loss of power by restricting the sample size. Given the distribution of missing data (S1 Table) and the large numbers offered by the registry, we felt that a complete case analysis was suitable for this study, and any reduction in power would be negligible. We cannot exclude the possibility that better surgeons may choose prostheses with lower revision rates.

The fact that the results seen at the RD&E were achieved with 83 different lead surgeons supports the theory that the implant is the driver of improved survival results rather than the skill of the individual surgeon. If this observed association is indeed true, the use of implants with evidence of good survival should be encouraged throughout the health service. Further work may focus on the effect of a change in implant use of a single hospital/unit on survival results in time-series analyses. Although implant selection appears to be associated with improved survival in THR, it is yet to be seen whether the same phenomenon is true in other branches of medicine heavily reliant on implantable devices. Further research in other areas is warranted to investigate this effect.

## Conclusion

In this study, we found evidence suggesting that implant selection is associated with the long-term survival of THRs rather than factors specific to a high-performing unit. Surgeons, commissioners, and patients should use this information when considering THR.

## Supporting information

**S1 STROBE Checklist. Annotated STROBE checklist detailing how this study meets the criteria laid out in the STROBE statement.** STROBE, Strengthening the Reporting of Observational Studies in Epidemiology.
(DOC)

**S1 Fig. Demonstration of goodness of fit of flexible parametric survival analysis model for total hip replacements using the Exeter V40 femoral stem.** Goodness of fit was assessed visually using the above figure as well as by assessment of log-likelihood of different models. A model with 5 knots was chosen as further knots provided more complexity with little improvement in log-likelihood.
(TIF)

**S2 Fig. Demonstration of nonproportionality of hazard of revision.** The above figure demonstrates that when using a flexible parametric survival analysis model that allows the hazard of failure to vary with time to compare TD and PH models. There is an apparent difference between the hazard of failure at the exemplar center and in all other units (the solid lines). The fact that these solid lines cross is highly indicative of the fact that the hazards are not proportional through the entire follow-up of the study. PH, proportional hazard; TD, time

dependent.
(TIF)

**S3 Fig. FPSA complete case analysis adjusted for age, sex, American Society of Anesthesiology score, and socioeconomic status.** Results presented for a 68-year-old female patient with an American Society of Anesthesiology score of 2 and in the 10th decile of IMD organized by LSOA. FPSA, flexible parametric survival analysis; IMD, Index of Multiple Deprivation; LSOA, Lower Layer Super Output Area.
(TIF)

**S4 Fig. FPSA complete case analysis adjusted for age, sex, American Society of Anesthesiology score, socioeconomic status, and body mass index.** Results presented for a 68-year-old female patient with an American Society of Anesthesiology score of 2 and in the 10th decile of IMD organized by LSOA and body mass index in World Health Organization category 2. FPSA, flexible parametric survival analysis; IMD, Index of Multiple Deprivation; LSOA, Lower Layer Super Output Area.
(TIF)

**S5 Fig. Comparison of the all-cause construct survival of the 5 most used cemented stems of all time in combination with any cup.** A comparison of the probability of all-cause revision (1 –Kaplan–Meier) for all constructs using the 5 most frequently implanted cemented femoral stems, demonstrating the differences in revision estimates between these stems. This suggests that the results demonstrated in this study may be achievable with other femoral stems.
(TIF)

**S1 Table. Distribution of BMI across categories.** A comparison of the distribution of BMI between the 2 exposure categories (Royal Devon & Exeter hospital and all other hospitals combined). BMI, body mass index.
(DOCX)

**S2 Table. Distribution of missing data.** Table detailing the distribution of missing data between the exposure categories (Royal Devon & Exeter hospital and all other hospitals combined).
(DOCX)

**S3 Table. At-risk table.** Table demonstrating the number of total hip replacements at risk at each time point following operation. For use in the interpretation of previous survival graphs.
(DOCX)

## Acknowledgments

We thank the patients and staff of all the hospitals in England, Wales and Northern Ireland who have contributed data to the National Joint Registry.

The Healthcare Quality Improvement Partnership (HQIP) and/or the NJR take no responsibility for the accuracy, currency, reliability, and correctness of any data used or referred to in this report, nor for the accuracy, currency, reliability, and correctness of links or references to other information sources and disclaims all warranties in relation to such data, links, and references to the maximum extent permitted by legislation.

HQIP and NJR shall have no liability (including but not limited to liability by reason of negligence) for any loss, damage, cost or expense incurred or arising by reason of any person using or relying on the data within this report and whether caused by reason of any error, omission, or misrepresentation in the report or otherwise. This report is not to be taken as

advice. Third parties using or relying on the data in this report do so at their own risk and will be responsible for making their own assessment and should verify all relevant representations, statements, and information with their own professional advisers.

The views expressed are those of the author(s) and not necessarily those of the NIHR or the Department of Health and Social Care.

## Author Contributions

**Conceptualization:** Jonathan Thomas Evans, Ashley William Blom, Andrew John Timperley, Matthew James Wilson, Adrian Sayers.

**Data curation:** Jonathan Thomas Evans, Adrian Sayers.

**Formal analysis:** Jonathan Thomas Evans, Adrian Sayers.

**Funding acquisition:** Jonathan Thomas Evans, Ashley William Blom, Andrew John Timperley, Adrian Sayers, Michael Richard Whitehouse.

**Investigation:** Jonathan Thomas Evans, Adrian Sayers, Michael Richard Whitehouse.

**Methodology:** Jonathan Thomas Evans, Ashley William Blom, Andrew John Timperley, Paul Dieppe, Matthew James Wilson, Adrian Sayers, Michael Richard Whitehouse.

**Project administration:** Jonathan Thomas Evans, Ashley William Blom.

**Resources:** Ashley William Blom.

**Supervision:** Ashley William Blom, Adrian Sayers, Michael Richard Whitehouse.

**Validation:** Jonathan Thomas Evans, Adrian Sayers.

**Visualization:** Jonathan Thomas Evans, Andrew John Timperley, Paul Dieppe, Matthew James Wilson, Adrian Sayers, Michael Richard Whitehouse.

**Writing – original draft:** Jonathan Thomas Evans, Paul Dieppe.

**Writing – review & editing:** Jonathan Thomas Evans, Ashley William Blom, Andrew John Timperley, Paul Dieppe, Matthew James Wilson, Adrian Sayers, Michael Richard Whitehouse.

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
