## [Editor Report · Decision Letter 0]

21 Jan 2020

Dear Dr Evans, 

Thank you for submitting your manuscript entitled "What makes a Total Hip Replacement last longer, the implant or who puts it in?

Findings from the National Joint Registry for England, Wales, Northern Ireland and the Isle of Man." for consideration by PLOS Medicine.

Your manuscript has now been evaluated by the PLOS Medicine editorial staff and I am writing to let you know that we would like to send your submission out for external peer review.

Kind regards,

Helen Howard, for Clare Stone PhD 

Acting Editor-in-Chief

PLOS Medicine 

plosmedicine.org

---

## [Decision Letter · Decision Letter 1]

26 Apr 2020

Dear Dr. Evans,

Thank you very much for submitting your manuscript "What makes a Total Hip Replacement last longer, the implant or who puts it in? Findings from the National Joint Registry for England, Wales, Northern Ireland and the Isle of Man." (PMEDICINE-D-20-00146R1) for consideration at PLOS Medicine. 

[LINK]

In light of these reviews, I am afraid that we will not be able to accept the manuscript for publication in the journal in its current form, but we would like to consider a revised version that addresses the reviewers' and editors' comments. Obviously we cannot make any decision about publication until we have seen the revised manuscript and your response, and we plan to seek re-review by one or more of the reviewers. 

We expect to receive your revised manuscript by May 15 2020 11:59PM. Please email us (plosmedicine@plos.org) if you have any questions or concerns.

We look forward to receiving your revised manuscript. 

Sincerely,

Emma Veitch, PhD

PLOS Medicine

On behalf of Clare Stone, PhD, Acting Chief Editor,

PLOS Medicine

plosmedicine.org

*We'd suggest using the general style for PLOS Medicine titles and therefore would simplify the second subpart of the title and just have "xxxyy: registry study" (rather than the current style which includes all the study settings).

*At this stage, we ask that you include a short, non-technical Author Summary of your research to make findings accessible to a wide audience that includes both scientists and non-scientists. The Author Summary should immediately follow the Abstract in your revised manuscript. This text is subject to editorial change and should be distinct from the scientific abstract. Please see our author guidelines for more information: https://journals.plos.org/plosmedicine/s/revising-your-manuscript#loc-author-summary

*In the abstract, the findings presented currently don't present effect sizes and confidence intervals so it's hard for the reader to get a sense of what the main findings really show (for the crude and adjusted analyses), particularly as the p-value for the adjusted analysis is not stat sig by conventional cutoffs. If possible, the effect estimates and CI's from the main text should also be added here together with the p-values that are currently given. 

*In the last sentence of the Abstract Methods and Findings section, please describe the main limitation(s) of the study's methodology.

*If possible, please change the in-text referencing style (which should be simple if referencing software was used) to giving the callouts in square brackets (eg, [1, 2] rather than superscript numbers). Many thanks

*Is it possible to clarify if the study had a prospective protocol or analysis plan? Please state this (either way) in the Methods section.

Comments from the reviewers:

Reviewer #1: Thanks for the opportunity to review your manuscript. My role is a statistical reviewer so I have concentrated on the data and analysis aspects of the manuscript, and the reporting of these. This manuscript aims to find whether individual or unit level factors lead to better outcomes in THR. I have provided overall comments, and then followed these with more specific queries. The page number refers to the page of the original word document not the compiled PDF. 

The main aim of this study is to disentangle individual (i.e. patient characteristics and implant types) from unit level factors. The unit effect (exemplar vs. other) is estimated as a fixed effect, without accounting for the multi-level structure of the data. For this type of research question a common approach would be to estimate a multi-level model for the outcome (a shared frailty model in this case of time-to-event data) and use the random intercepts to describe the variation, and then add individual and unit level factors to estimate whether accounting for composition (patient characteristic at sites) explains differences in the outcome. 

One of they key strengths of the manuscript is the large dataset from the joint replacement registry. My issues with the analysis come from the approach used to estimate unit vs individual effects and the missing data.

One of the fundamental parts of the study design is the selection of the RD&E unit as the reference unit for good survival outcomes for THRs. The last available report of the joint replacement registry highlighted several other units with better than expected outcomes for THR. It's not clear to me from this manuscript why just this individual unit was selected when there are apparently other high-performance units. Where these other units with good outcomes pooled with the rest of the units in the comparison of RD&E vs. other? 

The two major problems I can see with this approach is that it doesn't account for clustering within units (so the SE will be incorrect) and that there is no way to asses the variation across all of the units and come down to comparison of one selected unit with the rest in the registry.

The reporting of missing data isn't clear - ~30% of patients have missing BMI and presumably aren't included in the main analysis, but from Figure 1 I don't understand if these patients are excluded in this pathway, or if the exclusions used to create the complete case dataset are made after this point. It would be helpful for each of the covariates used in the analysis if a table reporting how many patients had missing data for each of the variables by the strata used to presented Table 1. 

A complete case analysis was used and justified as 'MI can introduce bias if not truly missing at random'. Unless done very poorly MAR approaches are also robust to when data is MCAR, and should be considered a more principled approach over complete-case analysis. The registry data does appear to have gaps in some of the variables that may make MI difficult to use (but given the description that exists it doesn't look impossible). I don't think this analysis necessarily needs to include MI, rather the limitations of a complete-case analysis should be adequately described in the discussion.

P4, L31. It's probably a result of the compiled PDF but Figure 1 looks like it has some image issues (it's a bit blurry).

P5, L1. Are there multiple patient records in this dataset (i.e. sequential unilateral HR)? 

P5, L22. What proportion of censored outcomes were administrative (end of data availability) and how many were for death? Death is a competing risk for implant failure and it seems likely that the censoring from death and failure are not independent. Is the proportion of censoring from death small enough it is unlikely change implant failure hazard rates accounting for competing risks?

P7, L26. BMI is included into the standard categories, but in my experience with patient data from joint replacement registries the number of patients with BMI >35 is much higher than seen in the general population. Is s the case for THRs in the UK? Does including the extended categories (particularly a 'super-obese' >40 category) provide additional adjustment that gives different results? 

Supplementary Fig1 + 2. For the Supplementary material, it would be helpful if there was an explanatory section for the graphical checks of non-proportionality and GOF - a brief explanation of what the diagnostic 

Reviewer #2: Thank you for the opportunity to review this work.

The work highlights the importance of informed and consistent surgeon choices rather than surgical skill in order to improve patient outcomes using NJR data. It demonstrates that the results achieved by a positive outlier centre are achievable by all surgeons rather than the elite few and uses robust data and methodology to do this.

The benefits of reducing variation and revision rates by adopting best practice are not just patient related but have far reaching cost implications on the health care system.

Using one centre that uses a single femoral stem and is a positive outlier in the NJR reduces bias and allows for this in-depth study, and is only achievable with analysis of registry data.

The use of flexible parametric survival analysis by the authors is commended, however, the majority of readers may not be familiar with this method. I would therefore suggest the authors expand on this method and why it is most suitable for this study. The use of sensitivity analysis and PPI also add to the validity if the findings.

In particular:

Methods:

Please expand on the suitability and use of FPSA.

Results:

In the adjusted analyses section, the restricted analysis is important as it is the construct that is the most important factor rather than the individual components and there are still acceptable numbers in this restricted analysis. Consider simplifying the sentence regarding the null hypothesis with regard to RMST as it is clunky and difficult to interpret, Please expand on the the difference in RMST between groups changing over time in Figure 4, as this may not be clear to some readers.

For the sensitivity analysis, the performance of the five most commonly implanted stems indicates that not all stems of the same mode of fixation are comparable. This is an important finding and may be lost in the supplementary material. Consider including in the main text or expanding on this finding.

Discussion:

The authors suggest that implant choice is responsible for the 'better than expected' results at the RD&E rather than surgeon skill. Does consistency also contribute to this and can this be measured? Are there any other centres that are as consistent in their implant choice (especially if it is a different implant) and are their results comparable? If surgeons consistently use the same implant, even if it is a poorer performing implant, do they have improved performance compared with less consistent surgeons?

In summary, this is an important study with a valuable message that has been examined using strong methodology with registry data. It identifies factors that are easily modifiable (i.e. implant choice) for a surgeon to improve implant survivorship and confirms the findings of the GIRFT initiative. It is concisely and clearly written, with appropriate methodology and conclusions, and provides novel information that is important within the discipline.

Reviewer #3: The authors address an important issue within the arthroplasty surgery community, what is the main driver for outcome (defined as revision) the implant or the surgeons (and his team) who puts it in.

The aim was to investigate whether "better than expected" survival results of one type of implant within one clinic was due to the unit or the implant.

For analysis they used Flexible parametric survival analysis (FPSA). The latter has the advantage of taken time-depended covariates into account, but this analysis technique may also show effects which may not be present. The authors assessed FPSA models for goodness of fit against KM curves.

The NJR data were complete 95% primary and 90% revision, what were those data for the RD&E unit?

What was the minimum number of TH procedures each surgeon did at the RD&E unit. For that matter in the discussion section it could be discussed that all surgeons performed at least x number, which also excludes a surgical factor being a confounder in the outcome "revision".

Strengths Weakness section

Please explore more e.g. why FPSA seems to better compared to KM with proportional hazard models ; competing risk analysis. But FPSA may have some disadvantage as well, it may "dictate" the survival curve as such. The latter will give more insight to the clinical reader on advantages and disadvantages of these two widely sued techniques. 

I would also recommend a statistician for review

Nevertheless, in conclusion I would congratulate the authors with their work

[LINK]

---

## [Decision Letter · Decision Letter 2]

2 Jul 2020

Dear Dr. Evans,

Thank you very much for re-submitting your manuscript "What makes a Total Hip Replacement last longer, the implant or who puts it in? A registry study" (PMEDICINE-D-20-00146R2) for review by PLOS Medicine.

I have discussed the paper with my colleagues and it was also seen again by one reviewer. I am pleased to say that provided the remaining editorial and production issues are dealt with we are planning to accept the paper for publication in the journal.

[LINK]

We look forward to receiving the revised manuscript by Jul 09 2020 11:59PM. 

Sincerely,

Caitlin Moyer, Ph.D.

Associate Editor 

PLOS Medicine

plosmedicine.org

Requests from Editors:

1.Data availability statement: Please provide the complete access information (a web address or contact email address) for access to the study data. 

2.Response to reviewer 3: Please do include the surgeon data description and mention in the discussion of how surgeon factors into your interpretation.

3.New reviewer 1 comments: Please do include a sentence in the manuscript regarding the issue of multiple implants from the same patient in the dataset being a rare risk, and please note in the Methods regarding the BMI categories that were used, as requested by the reviewer. 

4.Title: Please revise your title according to PLOS Medicine's style. Your title must be nondeclarative and not a question. It should begin with main concept if possible. Please place the study design ("A randomized controlled trial," "A retrospective study," "A modelling study," etc.) in the subtitle (ie, after a colon). Please also include the setting of the study in the title.

We suggest: “Factors associated with implant survival following total hip replacement surgery: A registry study of data from the National Joint Registry of England, Wales, Northern Ireland, and the Isle of Man” or similar.

5.Abstract: Background: Please revise the final sentence to “We aimed to investigate whether the statistically improved implant survival of THRs in a high performing unit is associated with the implants used or other factors at that unit, such as surgical skill.” to avoid implications of causality, which your study cannot address.

6.Abstract: Methods and Findings: Please revise the sentence “Either the selected “exemplar” unit or all others combined.” to clarify what this means.

7.Abstract: Methods and Findings: Please provide some information on the number of hospitals represented, and other summary demographic information to lend context here.

8.Abstract: Methods and Findings: For the second to last sentence, please revise to “A limitation is that this study is observational and conclusions regarding causality cannot be inferred.” or similar, to clarify.

9.Abstract: Conclusions: Please interpret the study based on the results presented in the abstract, emphasizing what is new without overstating your conclusions. For example, your conclusions speak to an association between performance and factors not mentioned in the Methods and Findings section of the abstract. Please revise the abstract so that the Conclusions are an interpretation of what you present in the Methods and Findings. 

In addition to clarifying this, we suggest revising the first sentence to: “Our results suggest that the “better than expected” performance of an exemplar centre is associated with implant choice rather than process factors such as surgical skill or experience.” to avoid implying causality.

10. Author Summary: Why Was This Study Done? Please combine bullet points to reduce the total number; we suggest:

--In general, Total Hip Replacement (THR) is safe and effective at reducing pain and restoring

mobility to people with end-stage arthritis of the hip.

--In England and Wales in 2017 over 822 different types of hip replacement were used, and different brands of hip replacement have been shown to have varying survival rates at different follow-up timepoints. Reducing variation in outcomes following surgery is an important aim of the National Health Service (NHS) in England and Wales.

--A national database of all hip replacements in England and Wales (the National Joint Registry) has shown variation in survival rates between different hospitals, and a few hospitals are highlighted by the database for having better survival rates than the others.

11. Author Summary: What Did the Researchers Do and Find?: Please add one bullet point describing the study (objective and method).

12. Author Summary: What do these findings mean?: We suggest the following revision:

--These findings suggest that the better results seen in this one hospital are not associated with the skill of the surgeon, or the set-up of the hospital but are associated with the choice of hip replacement.

--Future studies are needed to determine if this is also the case across other brands of hip replacement, and to determine whether the choice of implant is similarly associated with implant survival across other specialties.

13. Methods: Under “Sensitivity Analyses”: Please define the exact ranges used and provide the ranges for SES deciles and BMI WHO categories: “Socioeconomic status was assessed using deciles of the Index of Multiple Deprivation (IMD) organised by Lower-Layer Super Output Areas (LSOA) and BMI was treated as a categorical variable using World Health Organisation categories.” Please do include the BMI table provided in the response to reviewer 1, at least as supporting information.

14. Methods: Please provide the name(s) of the institutional review board(s) that provided ethical approval, and please indicate whether participant consent was obtained and the manner of consent (written or oral).

15. Methods: Please provide information on the patient data included in the NJR, the centers represented (region, etc.) and how participant data were obtained (please explain what is presented in Figure 1).

16. Results. The first sentence mentions “exclusions” and there does not seem to be any text here or in the methods describing how or why data were excluded from analysis. Please supply information describing excluded data.

17. Results: Adjusted analyses: Please refer to specific supplementary figures and tables, rather than generally referring to “supplementary material”

18. Discussion: Please present and organize the Discussion as follows: a short, clear summary of the article's findings; what the study adds to existing research and where and why the results may differ from previous research; strengths and limitations of the study; implications and next steps for research, clinical practice, and/or public policy; one-paragraph conclusion.

19. Please incorporate the section “Strengths and weaknesses of the study in relation to other studies” (“We are unaware of any studies to date investigating the reasons why one unit achieves better THR survival than others.”) into the rest of the discussion, this does not need to be an independent section.

20. Discussion: Unanswered questions and future research: Please revise this sentence to “Although implant selection appears to be associated with improved survival in THR, it is yet to be seen whether the same phenomenon is true in other branches of medicine heavily reliant on implantable devices. Further research in other areas is warranted to investigate this effect.” to remove implying causality. 

21. Discussion: Please expand on the discussion of unanswered questions and implications for clinical practice.

22. Conclusion: Please revise this sentence to: “In this study, we found evidence suggesting that implant selection is associated with the long-term survival of total hip replacements rather than factors specific to a high performing unit.” to avoid causal implications.

23. Please remove the sections: Contributions, Ethical Approval, Competing Interests, Transparency Statement, Role of the Funding Source, Disclaimer- and please make sure all of the information contained is submitted within the manuscript submission form.

24. References: References: Please place the in-text citation in brackets before the punctuation mark, like this [1].

Please use the "Vancouver" style for reference formatting, and see our website for other reference guidelines https://journals.plos.org/plosmedicine/s/submission-guidelines#loc-references

For example, please check reference 5, and 16, and please make sure journal abbreviations are consistent with Vancouver style.

25. Figure 2: Please define, in the legend, that the shaded region represents the 95% CI for each line. Please consider using different colors for the different lines, as it is difficult to differentiate the shades of gray. Please spell out “operation” or use something descriptive such as “THR” procedure, in the X axis label. Please use evenly spaced markers for the years on the X axis, unless there is a reason not to do so. Please define the abbreviation for THR and KM in the legend.

26. Figure 3: Please define THR, ASA and RD&E in the legend. Please spell out “operation” or use something descriptive such as “THR” procedure, in the X axis label. Please use evenly spaced markers for the years on the X axis, unless there is a reason not to do so. In the legend, please explain all the panels (3a, 3b, etc).

27. Figure 4: Please define RMST in the legend. Please define the shaded region as the 95% CIs. Please explain between which groups the difference is illustrating.

28. Figures: Please provide descriptive legends for all figures (including those in Supporting Information files).

29. Table 1: Please define abbreviations for NHS, BMI, THR, ASA in the legend. 

30. Supplementary material file: For all graphs please change “Years post op” to Years post operation or THR or similar. Please provide titles and legends for each individual table and figure in the Supporting Information (Figure 5, Table 1 and Table 2 have no legends), and make sure that all abbreviations used within figures and tables are spelled out in the legends.

31. Checklist: Please ensure that the study is reported according to the STROBE guideline, and include the completed STROBE checklist as Supporting Information. When completing the checklist, please use section and paragraph numbers, rather than page numbers. 

Please add the following statement, or similar, to the Methods: "This study is reported as per the Strengthening the Reporting of Observational Studies in Epidemiology (STROBE) guideline (S1 Checklist)."

Comments from Reviewers:

Reviewer #1: Thanks for the revised manuscript and detailed responses to my original queries. Overall, I think this manuscript is looking good with just a few suggested minor changes

With the research question further clarified, I agree with the authors that a multi-level model approach is not required for this study, with the focus on the type of implant rather than surgical unit. Being able fit the more sophisticated flexible survivals models is distinct advantage here so I am satisfied with the approach used. 

The information about THR and mortality was helpful thank you and being able to look at these references was reassuring. One of my original comments was about the possibility of multiple implants from the same patient in the dataset. If the risk is very rare then it is very unlikely to change the results by accounting for this, I would be happy if you could include a sentence explaining this with a reference.

The changes to the comments about missing data, and the clarifications about the exclusions around missing patients are now clear and I am happy with the changes. The additional information about BMI was useful for me but I agree that this doesn't necessarily need to be added to the manuscript if BMI was only used in one of the sensitivity analyses. A note in the methods that you have used these extended categories (relative the usual <25, 25-<30, 30+ which most people try to get away with) would be helpful.

The revised supplementary material is helpful and I think this should be a helpful for a general audience who want to understand the flexible survival analysis.

I agree with the comments you made in response to reviewer 3 - the FPS should allow the curve to better fit the data and even if the data showed PHs it would still give a valid result (and one similar to Cox in that situation). 

The question you raise in the last sentence of the strengths and weaknesses section is a good one - next project? (no changes required, just a comment)

[LINK]

---

## [Editor Report · Decision Letter 3]

30 Jul 2020

Dear Mr Evans, 

On behalf of my colleagues and the academic editor, Dr. Rob G.H.H. Nelissen, I am delighted to inform you that your manuscript entitled "Factors associated with implant survival following total hip replacement surgery: A registry study of data from the National Joint Registry of England, Wales, Northern Ireland, and the Isle of Man" (PMEDICINE-D-20-00146R3) has been accepted for publication in PLOS Medicine. 

PRODUCTION PROCESS

PRESS

PROFILE INFORMATION

Thank you again for submitting the manuscript to PLOS Medicine. We look forward to publishing it. 

Best wishes, 

Caitlin Moyer, Ph.D.

Associate Editor 

PLOS Medicine

plosmedicine.org